# Mechanism of MRX inhibition by Rif2 at telomeres

Florian Roisné-Hamelin[1], Sabrina Pobiega[1], Kévin Jézéquel[1], Simona Miron[2], Jordane Dépagne[3], Xavier Veaute [3], Didier Busso [3], Marie-Hélène Le Du[2], Isabelle Callebaut[4], Jean-Baptiste Charbonnier[2], Philippe Cuniasse[2], Sophie Zinn-Justin[2] & Stéphane Marcand [1✉]

Specific proteins present at telomeres ensure chromosome end stability, in large part through unknown mechanisms. In this work, we address how the *Saccharomyces cerevisiae* ORC-related Rif2 protein protects telomere. We show that the small N-terminal Rif2 BAT motif (Blocks Addition of Telomeres) previously known to limit telomere elongation and Tel1 activity is also sufficient to block NHEJ and 5' end resection. The BAT motif inhibits the ability of the Mre11-Rad50-Xrs2 complex (MRX) to capture DNA ends. It acts through a direct contact with Rad50 ATP-binding Head domains. Through genetic approaches guided by structural predictions, we identify residues at the surface of Rad50 that are essential for the interaction with Rif2 and its inhibition. Finally, a docking model predicts how BAT binding could specifically destabilise the DNA-bound state of the MRX complex. From these results, we propose that when an MRX complex approaches a telomere, the Rif2 BAT motif binds MRX Head in its ATP-bound resting state. This antagonises MRX transition to its DNA-bound state, and favours a rapid return to the ATP-bound state. Unable to stably capture the telomere end, the MRX complex cannot proceed with the subsequent steps of NHEJ, Tel1-activation and 5' resection.

[1] Université de Paris, Université Paris-Saclay, Inserm, CEA, Institut de Biologie François Jacob, iRCM, UMR Stabilité Génétique Cellules Souches et Radiations, Fontenay-aux-Roses, France. [2] Université Paris-Saclay, CNRS, CEA, Institute for Integrative Biology of the Cell (I2BC), Gif-sur-Yvette, France. [3] CIGEx, Université de Paris, Université Paris-Saclay, Inserm, CEA, Institut de Biologie François Jacob, iRCM, UMR Stabilité Génétique Cellules Souches et Radiations, Fontenay-aux-Roses, France. [4] Sorbonne Université, Muséum National d'Histoire Naturelle, UMR CNRS 7590, Institut de Minéralogie de Physique des Matériaux et de Cosmochimie (IMPMC), Paris, France. ✉email: stephane.marcand@cea.fr

Telomeres are protein–DNA complexes ensuring that native chromosome ends escape the pathways acting on broken DNA ends[1–5]. The repressed pathways are non-homologous end joining (NHEJ), 5′ end resection, homologous recombination and the DNA damage checkpoint. In addition, telomeres in association with telomerase solve the problem of replicating chromosome ends by semiconservative DNA replication. They also control telomere length homoeostasis to avoid the occurrence of excessively short or long telomeres. These core telomere functions are established by a relatively small number of proteins specifically present or enriched at telomeres. How each factor acts at a molecular level is only partially deciphered[6–19].

Telomere proteins are usually not restricted to a unique telomeric function and can control several pathways. In this regard, the budding yeast telomere factor Rif2 is paradigmatic. Rif2 is a globular 46 kDa protein with a single folded AAA+ domain originating from a duplication of the *ORC4* gene[20–22] (Fig. 1A, Supplementary Fig. 1). It is recruited to telomeres by Rap1, the protein covering telomere sequences in budding yeast species[22,23]. Rif2 interacts with Rap1 C-terminal domain through two distinct interfaces: the Rif2 AAA+ domain and a small Rap1 Binding Motif (RBM) in N-terminal position (residues 37–48)[22]. Both epitopes interact with Rap1 with similar affinities ($i_d$ ~30–50 μM) and synergise to ensure an efficient Rif2 recruitment to telomeres. They may also allow Rif2 molecules to interconnect adjacent telomere-bound Rap1 molecules[22]. Rif2 contributes to several telomere functions. Rif2 limits telomere elongation by telomerase[23–30]. It also represses NHEJ, preventing telomere-telomere fusions[21,31] and inhibits 5′ end resection, homologous recombination and checkpoint activation[27,32–37].

A shared feature of these pathways targeted by Rif2 is the involvement of the Mre11–Rad50–Xrs2[NBS1] complex (MRX[MRN]), an ATPase related to the SMC family (Structural Maintenance of Chromosome). At double-strand breaks, the MRX[MRN] complex is recruited early to the broken ends where it has multiple roles. First, it promotes NHEJ repair through tethering of the broken ends and through interactions with the NHEJ factors KU, Lif1[XRCC4] and Nej1[XLF38–44]. In contexts where NHEJ does not occur, MRX[MRN] promotes 5′ end resection, Mec1[ATR/Rad3] checkpoint kinase activation and repair by homologous recombination through its nuclease activities[45,46] and through interactions with chromatin remodellers and regulation of long-range resection actors[47–49]. At telomeres, MRX[MRN] is also essential to Tel1[ATM] kinase recruitment and activation, a key factor for chromosome ends maintenance by telomerase[50–54].

The intersection between Rif2 and MRX functions suggests that the MRX complex is a Rif2 target at telomeres. Supporting this model, Rif2 inhibits the recruitment of the MRX complex to DNA ends either directly or by inhibiting Tel1, which stabilises the complex interaction with DNA[29,33,55].

Previous studies addressing Rif2 molecular mechanisms focused on the inhibition of Tel1 and telomere elongation. This function of Rif2 involves a small BAT motif (Blocks Addition of Telomeres) present in the N-terminal position (residues 1–36) just upstream of the RBM motif[28] (Fig. 1A). Targeting the BAT to telomeres through a covalent fusion with Rap1 shortens telomeres, even in the absence of full-length Rif2 or Rap1 C-terminal domain. This shows that the Rif2 BAT motif is sufficient to inhibit telomere elongation[28]. In vitro Rif2 represses DNA/MRX-dependent activation of Tel1, indicating that it can directly act on the MRX–Tel1 complex to inhibit telomere elongation[30]. Rif2 N-terminal region was initially proposed to interact with the Xrs2 C-terminal motif that binds to Tel1, therefore antagonising Tel1 recruitment and function at telomeres[33]. However, telomere shortening by the BAT still occurs in the absence of this Xrs2 C-terminal motif[28]. Furthermore, Rif2 N-terminal region interacts

in vitro with Rad50 and stimulates its ATPase activity independently of Xrs2[29,30], suggesting that Rif2 acts on Rad50, and not on Xrs2, to repress telomere elongation.

Here we asked whether the function of the Rif2 BAT motif is restricted to telomere elongation inhibition. We found that the BAT is sufficient to inhibit NHEJ, 5′ resection and the stable interaction of the MRX complex with DNA ends. We showed that the BAT interacts with Rad50 ATP-binding domains. We defined a minimal active motif of 26 residues sufficient for both interaction and function, and showed that Rif2 action through the BAT is restricted to short distances *in cis*. Through genetic approaches guided by structural predictions, we identified residues at the surface of Rad50 that are essential for the interaction with Rif2 and for its inhibition. The position of these residues on Rad50 leads us to propose that Rif2 opposes MRX complex functions by precluding the formation of its DNA-bound active state.

## Results

**Rif2 N-terminal region inhibits chromosome end fusions.** Targeting the Rif2 BAT motif to telomeres represses telomere elongation[28]. Can it also protect telomeres against the NHEJ pathway? In budding yeast, Rap1, Rif2 and another Rap1-interacting factor, Sir4, act in synergy to prevent NHEJ-dependent chromosome ends fusions[21,31]. The loss of both Rif2 and Sir4 is needed to result in telomere fusions frequent enough to be efficiently detected by PCR (Fig. 1B left panel, telomeres being heterogeneous in length, amplified telomere fusions appear as a smeared signal). Canonical NHEJ is the sole pathway producing these fusions since their occurrence requires Lif1[XRCC4], an essential co-factor of Lig4 (Fig. 1B, compare *rif2Δ sir4Δ* with *rif2Δ sir4Δ lif1Δ*)[21]. To test BAT ability to block NHEJ, we targeted Rif2 N-terminal region (1–60) to telomeres by fusing it to endogenous Rap1 C-terminal end. A 10-Glycine linker connects the two sequences (chimera from[28], referred here as *RAP1-RIF2$_{1-60}$*). As expected[28], fusing the BAT to Rap1 shortens telomeres (Supplementary Fig. 2A). More strikingly, it also inhibits the occurrence of chromosome ends fusions in cells lacking Rif2 and Sir4, bringing it back to levels observed in *RIF2+* cells devoid of Sir4 (Fig. 1B left panel). Thus, targeting the Rif2 N-terminal region to telomeres is sufficient to inhibit NHEJ in the absence of the full-length protein. We also observed this inhibition in cells lacking Tel1, where telomeres are about half shorter and fusions more frequent (Fig. 1B right panel)[21]. This indicates that the Rif2 N-terminal region, in the same way as the full-length protein, can bypass Tel1 to repress NHEJ at telomeres.

**Rif2 BAT motif inhibits NHEJ at broken ends.** Next, we asked whether the Rif2 N-terminal region and more specifically the BAT motif can act at another location than telomeres. To assay NHEJ activity we used as a proxy survival to double-strand breaks (DSBs) induced by the continuously expressed I-SceI endonuclease (Fig. 2A, Supplementary Fig. 2B) (see figure legend for details)[21,56,57]. In these assays, Gal4 binding sites are present next to a I-SceI cutting site. Fusing peptides to Gal4 DNA binding domain (Gal4$_{DBD}$) allows us to target them at a double-strand end to test their impact on NHEJ efficiency. Targeting full-length Rif2 through Rap1 C-terminal domain (Rap1$_{Cter}$) inhibits NHEJ in these assays[21]. Targeting Rif2 N-terminal region (1-60) or a smaller Rif2 peptide only including the BAT motif, not the RBM (1-36) also lowers NHEJ-dependent survival relative to control situations where unfused Gal4$_{DBD}$ is bound to the Gal4 sites and where Gal4$_{DBD}$-Rif2$_{1-36/60}$ proteins remain untargeted to broken end (0 Gal4 site) (Fig. 2B, Supplementary Fig. 2B). NHEJ inhibition by the BAT still takes place in cells lacking endogenous

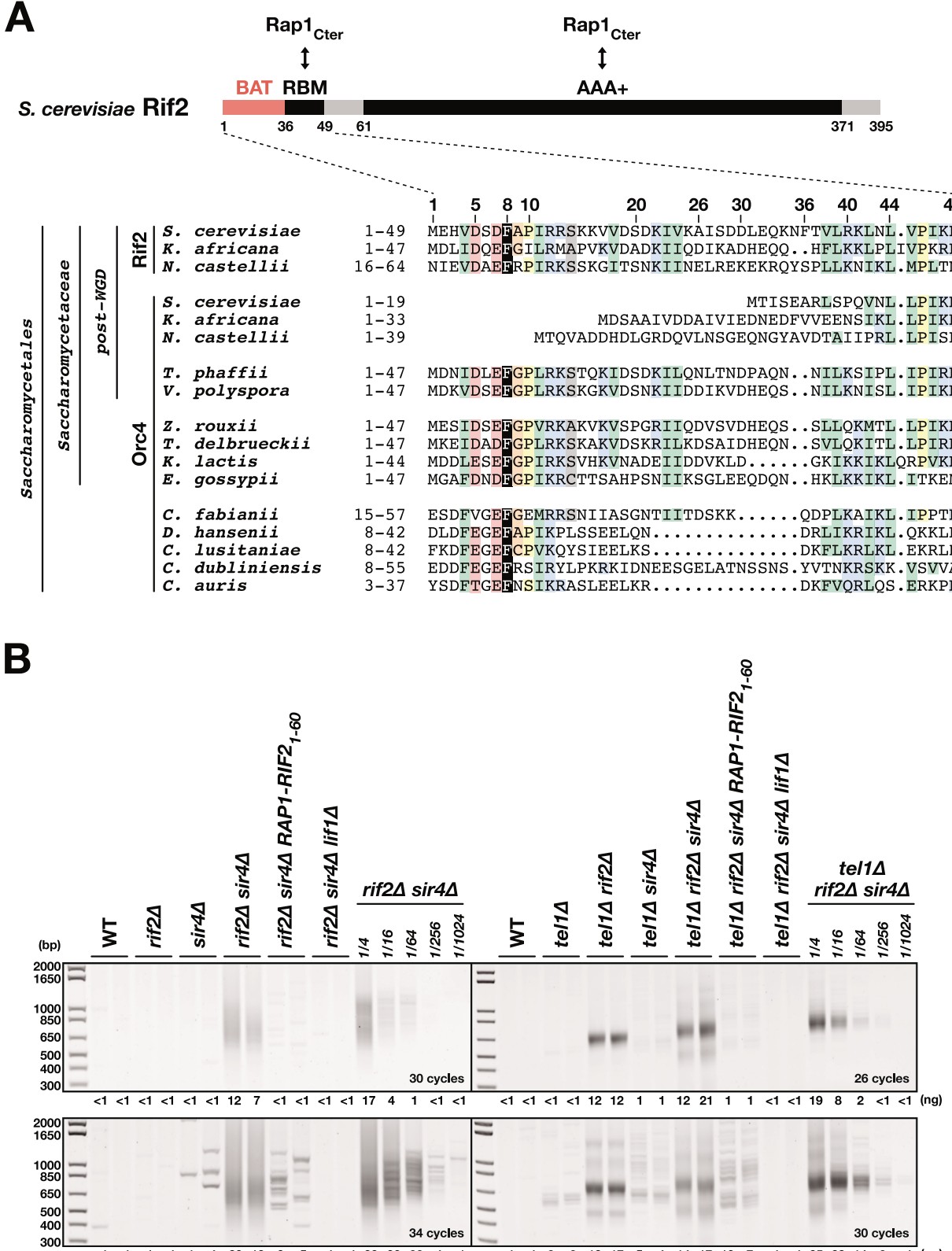

Rif2 (Fig. 2B) or Rap1 C-terminal domain (Supplementary Fig. 2C). This indicates that in these assays the BAT ability to repress NHEJ does not rely on the endogenous Rif2 protein nor on the Rap1 domain that recruits Rif2 to telomeres. The targeting of a single Gal4$_{DBD}$-Rif2$_{1-36}$ dimer inhibits NHEJ (1 Gal4 site in Fig. 2B and Supplementary Fig. 2B), showing that one or two BAT molecules are sufficient to act.

In the previous experiment, the edge of the Gal4 sites is 23 bp away from the broken end. To assess how far Rif2 BAT can act, we inserted sequences between the Gal4 sites and the I-SceI site. Strikingly, BAT ability to inhibit NHEJ rapidly weakens with increasing distances from the broken end (Fig. 2C, Supplementary Fig. 2D). Distances of 79 bp or more prevent BAT action. A similar threshold is found when NHEJ is inhibited by full-length

**Fig. 1 Rif2 N-terminal region inhibits NHEJ at telomere. A** Schematic representation of *S. cerevisiae* Rif2 and sequence conservation of Rif2 and Orc4 N-terminal region in *Saccharomycetales* species[21,22,28,30]. In some post-WGD (whole genome duplication) *Saccharomycetacae* species, Rif2 and Orc4 are syntenic and only Rif2 possesses a BAT motif. In other *Saccharomycetacae* species, the motif is found in Orc4. Core residues of the BAT motif are also present in Orc4 N-terminal region in non-*Saccharomycetacae* species of the *Saccharomycetales* order[30]. The alignment was extended downstream towards a conserved motif, which might correspond to a Rap1-binding module (RBM)[22]. Alignment of the full-length proteins is shown in Supplementary Fig. 1. (Sequence Accession numbers: Rif2**:** *Saccharomyces cerevisiae*: Q06208, *Kazachstania africana*: XP_003954885.1, *Naumovozyma_castellii*: XP_003677686.1, Orc4**:** *Saccharomyces cerevisiae*: P54791, *Kazachstania africana*: XP_003955485.1, *Naumovozyma_castellii*: XP_003674441.1, *Tetrapisispora_phaffii*: XP_003686307.1, *Vanderwaltozyma_polyspora*: XP_001643535.1, *Zygosaccharomyces_rouxii*:XP_002496500.1, *Torulaspora_delbrueckii*: XP_003683125.1, *Kluyveromyces_lactis*: XP_452959.1, *Eremothecium_gossypii*: NP_983126.1, *Cyberlindnera fabianii*: ONH65289.1, *Debaryomyces hansenii*: XP_459748.2, *Clavispora XP*_002615148.1, *Candida dubliniensis* XP_002420502.1, *Candida auris*: PIS52278.1). **B** Fusing Rif2 N-terminal region to Rap1 C-terminal end (*RAP1-RIF2$_{1-60}$*) protects telomeres against NHEJ-dependent fusions in cells lacking Rif2 and Sir4. Fusions between X and Y′ telomeres were detected by semi-quantitative PCR (upper panels: 30 and 26 cycles, lower panels: 34 and 30 cycles). Quantification of the amplified products indicated for each lane (ng). Serial 4-fold dilution of the template DNA from *rif2Δ sir4Δ* and *tel1Δ rif2Δ sir4Δ* cells provides an estimation of the method sensitivity. Rarer fusions are amplified as discrete bands. Experiment reproduced three times.

Rif2 targeted through Rap1 C-terminal domain. In addition, Rif2 overexpression does not inhibit NHEJ when untargeted to DNA ends (Supplementary Fig. 3A&B). These results show that Rif2 and Rif2 BAT need to be actively concentrated in the immediate vicinity of the broken end to block NHEJ.

Next, we determined if smaller N-terminal Rif2 fragments retain the ability to inhibit NHEJ. C-terminal truncations down to position 26 have no significant effect (Fig. 2D, 23 bp between the edge of the Gal4 sites and the broken end). More severe C-terminal truncations (1–24, 1–20) and N-terminal truncations (6–31) reduce or abolish BAT function. Thus, the first 26 residues of Rif2 are sufficient to inhibit NHEJ when targeted to a broken end.

To better establish the specificity of BAT impact on NHEJ, we mutated conserved residues within the *S. cerevisiae* Rif2 N-terminal region (1–60) (Fig. 1A) and targeted these mutated fragments to a DSB (Fig. 2E). This mutagenesis identifies F8, R12, R13, S14 and I23 as important residues for NHEJ inhibition. Mutations altering the Rap1 Binding Motif (R40A and L44A) have no significant effect (Fig. 2E). Then we tested the role of F8, the most highly conserved residue, in the context of the full-length Rif2 protein. F8 is essential for NHEJ inhibition, both at a broken end when it is ectopically targeted there (1–395 vs 1–395 *F8A*) (Fig. 2E) and at telomeres through its native recruitment by Rap1 (Fig. 2F). As previously observed[28], the *F8A* mutation elongates telomeres and does no impact protein stability (Supplementary Fig. 3C).

In addition to this mutagenesis, we asked if a Rif2 homologue from a distinct yeast species inhibits NHEJ. In *S. cerevisiae* and some other post-WGD (whole genome duplication) *Saccharomycetaceae* species, Rif2 and Orc4 are orthologs and only Rif2 possesses the BAT motif[20,21,30] (Fig. 1A, Supplementary Fig. 1). In other *Saccharomycetaceae* species, including *Kluyveromyces lactis*, there is only one protein, Orc4, and it possesses both the BAT motif and the Rap1-binding motif. We overexpressed *K. lactis* Orc4 in *S. cerevisiae* cells lacking Rif2. This restores Rap1 ability to inhibit NHEJ at a broken end (Fig. 2G) and at telomeres (Supplementary Fig. 3D). As in *S. cerevisiae* Rif2, *K. lactis* Orc4 F8 is essential to this inhibition (Fig. 2G). Together, these results show that the conserved BAT motif blocks NHEJ.

**Rif2 BAT blocks 5′ resection and inhibits MRX binding to broken ends**. Since Rif2 antagonizes 5′ DNA end resection[29,34–36], we asked if the BAT motif would be sufficient for this function too. To monitor resection, we used the I-SceI/Gal4 assay described above and a Southern blot approach (Supplementary Fig. 4A). Cells were arrested in G1 or in G2/M prior to induction of the I-SceI endonuclease. Targeting the Rif2 N-terminal region (1–60 and 1–36) (Fig. 3A, Supplementary Fig. 4B, C) or the Rap1 C-terminal domain (Supplementary Fig. 4C)

stabilises the adjacent broken end relative to the control conditions (empty vector or Gal4$_{DBD}$ alone). These results show that Rif2 BAT limits 5′ end resection both in G1 and G2/M phases.

Since Rif2 inhibits the interaction of MRX with DNA ends[29,33,55], we next tested BAT ability to oppose this interaction using a ChIP approach. As shown in Fig. 3B, targeting Rif2 N-terminal region (1–36) limits Mre11 and Xrs2 presence at the adjacent broken end relative to the control condition (Gal4$_{DBD}$ alone). Since MRX functionality requires its three subunits, this result shows that the BAT motif antagonises MRX stable interaction with DNA ends. This property can explain its ability to inhibit telomere elongation[28], NHEJ (Figs. 1, 2) and 5′ end resection (Fig. 3A).

**Rif2 BAT interacts with Rad50 ATP-binding domains**. Next, we addressed the question of how the BAT motif opposes the MRX complex at DNA ends by searching for proteins interacting with the BAT. In vitro, Rif2 interacts with Rad50[29,30]. We used a Two-Hybrid approach to test for interactions with each full-length subunit of the MRX complex (Fig. 4, Supplementary Fig. 5). We found that Rif2 N-terminal region (1–60 and 1–36) interacts with Rad50. This interaction does not require Mre11 and Xrs2 (Fig. 4A).

To assess the significance of this interaction between the BAT motif and Rad50, we tested its sensitivity to mutations that impact BAT ability to inhibit NHEJ. Mutations in the F8 and R12 residues, C-terminal truncation past residue 26 and a N-terminal truncation (6-31) weaken or abolish the interaction with Rad50 (Fig. 4B). These data show a good correlation between the strength of NHEJ inhibition by Rif2 N-terminal region (Fig. 2D) and its ability to interact with Rad50 in vivo. Interestingly, Rif2$_{1-60}$ F8A and R12A fragments still interact with Rap1. By contrast, mutation L44A in the Rap1-binding motif prevents the interaction with Rap1, as expected[22] but does not impact the interaction with Rad50. This further shows that Rif2 Rad50-binding motif (BAT) and Rap1-binding motif (RBM) are distinct and functionally separable.

Next, we asked which domain of Rad50 interacts with the BAT motif. Rad50 is an SMC protein with a fold-back structure[58–62]. Its N-terminal and C-terminal domains interact to form a globular ATPase Head. The rest of the protein assembles into a long coiled-coil with a median Zn-hook. Our attempt to arbitrarily truncate the protein into distinct domains failed in the Two-Hybrid assay. For this reason, we screened a library of random truncations within the Rad50 coiled-coil region and selected for clones still interacting with the BAT motif in Two-Hybrid. The shortest clone we obtained is a truncation of the coiled-coil region that connects the N-terminal and C-terminal Head domains (Δ190–1124, hereafter Rad50$_{\Delta cc}$) (Fig. 4C). Its interaction with Rif2 BAT does not require the endogenous full-length Rad50 and remains

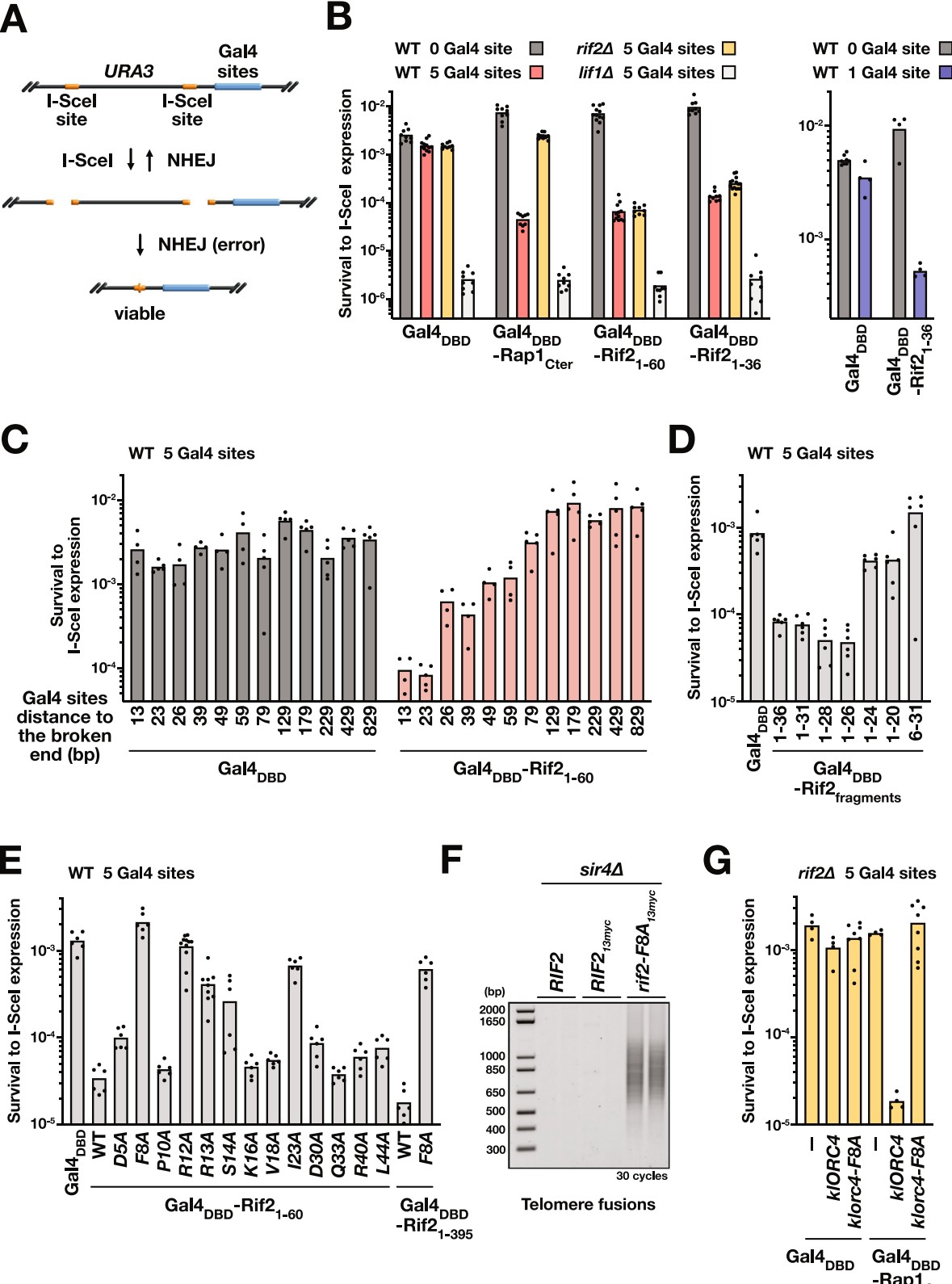

**Fig. 2 Rif2 BAT motif inhibits NHEJ at broken ends. A** I-SceI assay used to estimate NHEJ efficiency. Two inverted I-SceI sites are inserted at the endogenous *URA3* gene. Most survivors to continuous I-SceI expression have eliminated the I-SceI sites by fusing the distal broken ends[21]. **B** NHEJ inhibition by Rap1 C-terminal domain and Rif2 N-terminal region targeted at broken ends (*lif1Δ*: NHEJ-deficient cells). Means from independent cell cultures. **C** Increasing distances between the broken end and the Gal4 binding sites decrease NHEJ inhibition by Rif2. **D** Rif2 N-terminal truncations impacting the ability to inhibit NHEJ at broken ends. Gal4$_{DBD}$ and Gal4$_{DBD}$-Rif2$_{fragments}$ expressed from a centromeric plasmid. **E** Rif2 mutations impacting its ability to inhibit NHEJ at broken ends. Gal4$_{DBD}$ and Gal4$_{DBD}$-Rif2$_{1-60}$ expressed from an integrated plasmid. Gal4$_{DBD}$-Rif2$_{1-395}$ expressed from a centromeric plasmid. **F** The *rif2-F8A* mutation exposes telomeres to NHEJ in cells lacking Sir4 (fusions between X and Y' telomeres). Experiment reproduced three times. **G** *K. lactis* Orc4 expression complements Rif2 loss for NHEJ inhibition by Rap1 C-terminal domain in *S. cerevisiae*.

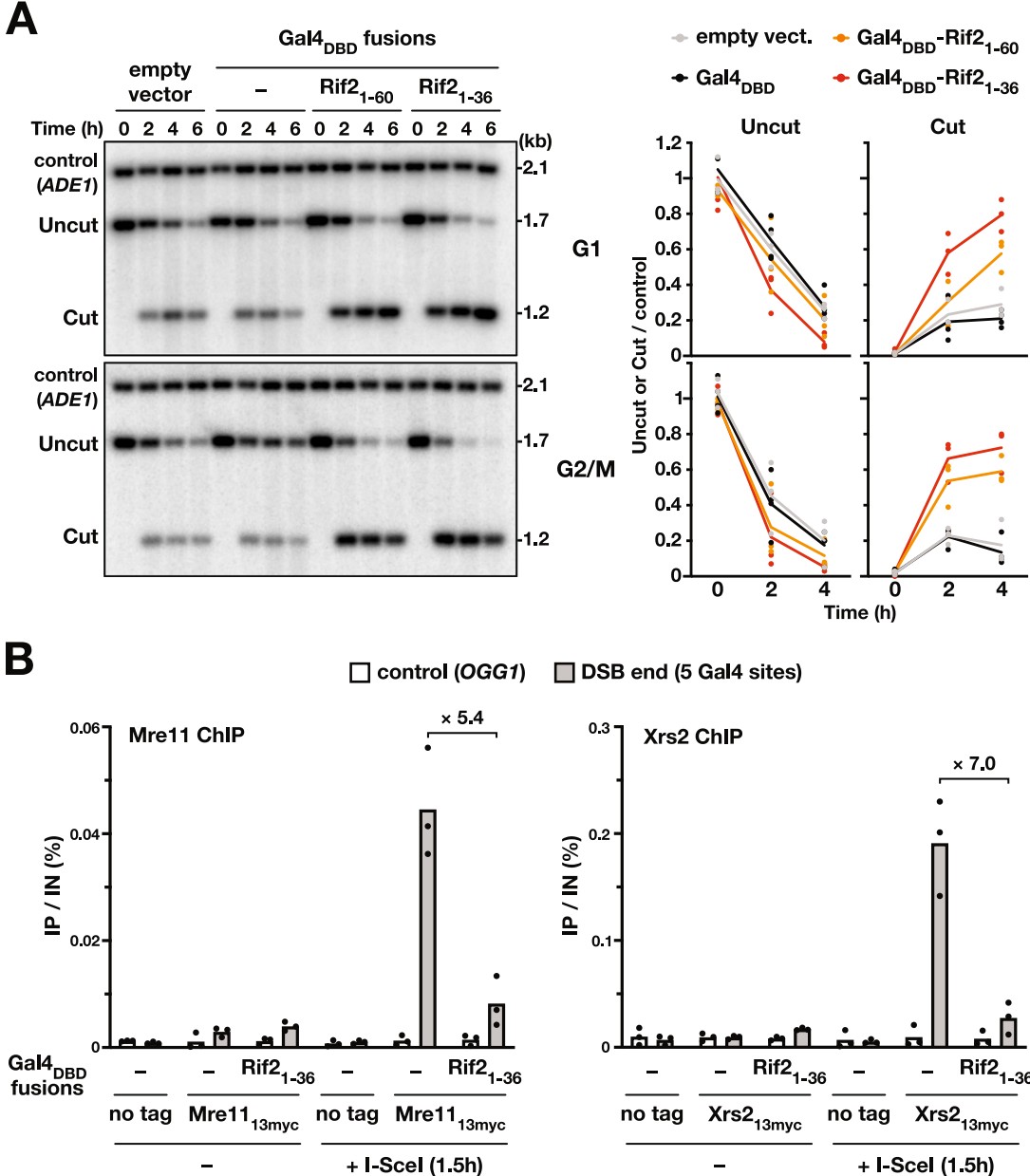

**Fig. 3 Rif2 BAT blocks 5′ resection and MRX complex presence at broken ends. A** Rif2 N-terminal region stabilises broken ends. Left panel: the stability of I-SceI-induced broken ends with 5 Gal4 sites determined by Southern blot in G1 and G2/M arrested cells. Right panel: quantification of the uncut and cut signals normalised to the control *ADE1* signal. Means from independent samples. Experiment reproduced three times. **B** Mre11 and Xrs2 presence at I-SceI-induced broken ends with 5 Gal4 sites determined by ChIP in G1 arrested cells. Quantification of immunoprecipitated DNA (IP) relative to the input DNA (IN). Means from independent samples. Quantification of I-SceI cleavage efficiency in Supplementary Fig. 4D.

sensitive to the F8A mutation (Fig. 4D). The interaction between Rad50 Head domains and Rif2₁₋₆₀ can also be observed in vitro using a GST pull-down assay (Fig. 4E).

**Identification of Rif2-resistant Rad50 mutants through a genetic screen.** Rif2 BAT mutants defective for NHEJ inhibition are also defective for interaction with Rad50. Therefore, we expect Rad50 mutants specifically defective for interaction with Rif2 to be insensitive to Rif2 BAT inhibition. Finding such mutants will inform us on where the BAT motif interacts on Rad50, information that may give insights into the mechanism of inhibition. To this end, we first performed a genetic screen to identify Rad50 mutants that would be prone to telomere fusions, that is to say defective for telomere protection by Rif2.

We used a genetic assay capable of capturing and quantifying chromosome fusions in budding yeast. This assay relies on the controlled inactivation of one centromere (*CEN6*) to select and rescue unstable dicentric chromosome fusions (Fig. 5A)[63]. Survival to centromere loss correlates with the frequency of chromosome fusions. Survival is low in wild-type cells (≈10⁻⁷ events/cell) and increases in cells lacking Rif2 and/or Sir4. To sensitise the assay, we choose to screen Rad50 mutants in cells lacking Sir4, where telomere protection relies more on Rif2 (Fig. 5A). Rad50 random PCR mutagenesis was performed on the sequence encoding the N-terminal part of the protein. We tested 300 mutants for survival to centromere loss. Among them, 13 increase the frequency of survival (referred as *m1* to *m13*) (Fig. 5B).

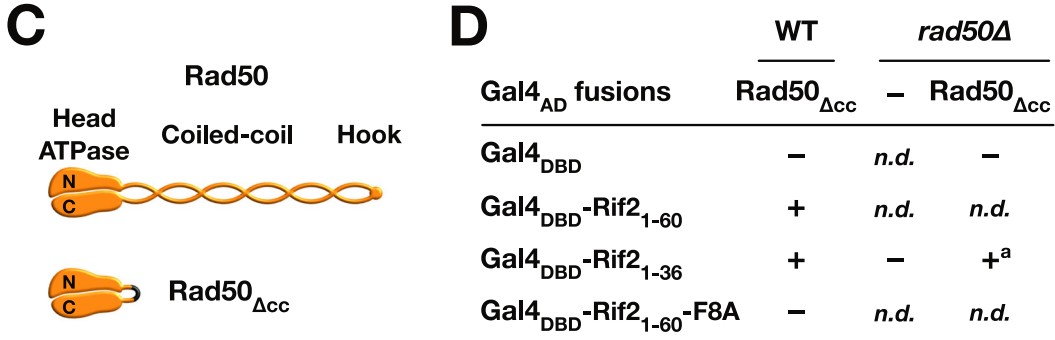

**A**

| Gal4$_{AD}$ fusions | WT | | mre11Δ | | xrs2Δ | |
|---|---|---|---|---|---|---|
| | – | Rad50 | – | Rad50 | – | Rad50 |
| Gal4$_{DBD}$ | – | – | *n.d.* | – | *n.d.* | – |
| Gal4$_{DBD}$-Rif2$_{1-60}$ | – | + | – | +[a] | – | +[a] |
| Gal4$_{DBD}$-Rif2$_{1-36}$ | – | + | – | +[a] | – | +[a] |

**B**

| Gal4$_{DBD}$ fusions | Rif2$_{1-60}$ | | | | Rif2$_{Nter-fragments}$ | | | | | | |
|---|---|---|---|---|---|---|---|---|---|---|---|
| | WT | F8A | R12A | L44A | 1-36 | 1-31 | 1-28 | 1-26 | 1-24 | 1-20 | 6-31 |
| Gal4$_{AD}$ | – | – | – | – | – | – | – | – | – | – | – |
| Gal4$_{AD}$-Rad50 | + | – | – | + | + | + | + | + | +/- | +/- | – |
| Gal4$_{AD}$-Rap1 | + | + | + | – | – | – | – | – | – | – | – |

**C**

Rad50
Head ATPase — Coiled-coil — Hook

Rad50$_{Δcc}$

**D**

| Gal4$_{AD}$ fusions | WT | rad50Δ | |
|---|---|---|---|
| | Rad50$_{Δcc}$ | – | Rad50$_{Δcc}$ |
| Gal4$_{DBD}$ | – | *n.d.* | – |
| Gal4$_{DBD}$-Rif2$_{1-60}$ | + | *n.d.* | *n.d.* |
| Gal4$_{DBD}$-Rif2$_{1-36}$ | + | – | +[a] |
| Gal4$_{DBD}$-Rif2$_{1-60}$-F8A | – | *n.d.* | *n.d.* |

**E**

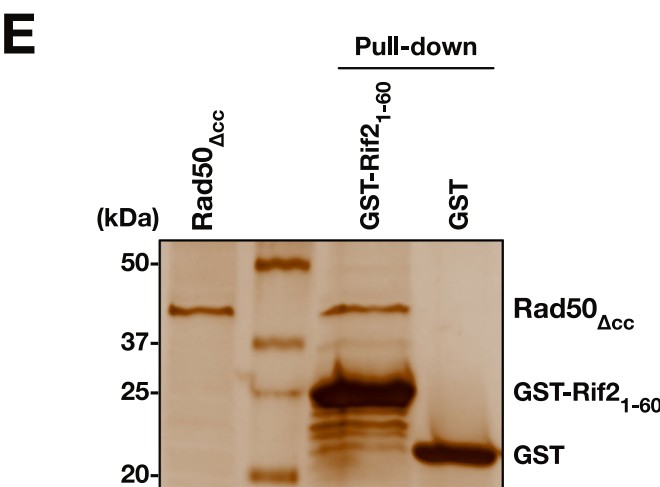

Pull-down

**Fig. 4 Rif2 BAT interacts with Rad50 ATPase Heads in vivo. A** 2-Hybrid interactions between Rif2 N-terminal region and full-length Rad50 in WT cells and in cells lacking Mre11 and Xrs2 (− no growth on plates supplemented with 3-AT, + growth on plates supplemented with 3-AT). **A**: slow growth in MRX-defective cells. **B** 2-Hybrid interactions between Rif2 N-terminal region and full-length Rad50 or Rap1 (fragment 366–827) in WT cells (+/- slow growth on plates supplemented with 3-AT). **C** Representation of full-length Rad50 and of the Rad50$_{ΔCC}$ fragment lacking the coiled-coil arm. **D** 2-Hybrid interactions between Rif2 N-terminal region and Rad50 ATPase Head in WT cells and in cells lacking the endogenous Rad50. **E** GST pull-down interaction between Rif2 N-terminal region and Rad50 ATPase Head. Experiment reproduced three times.

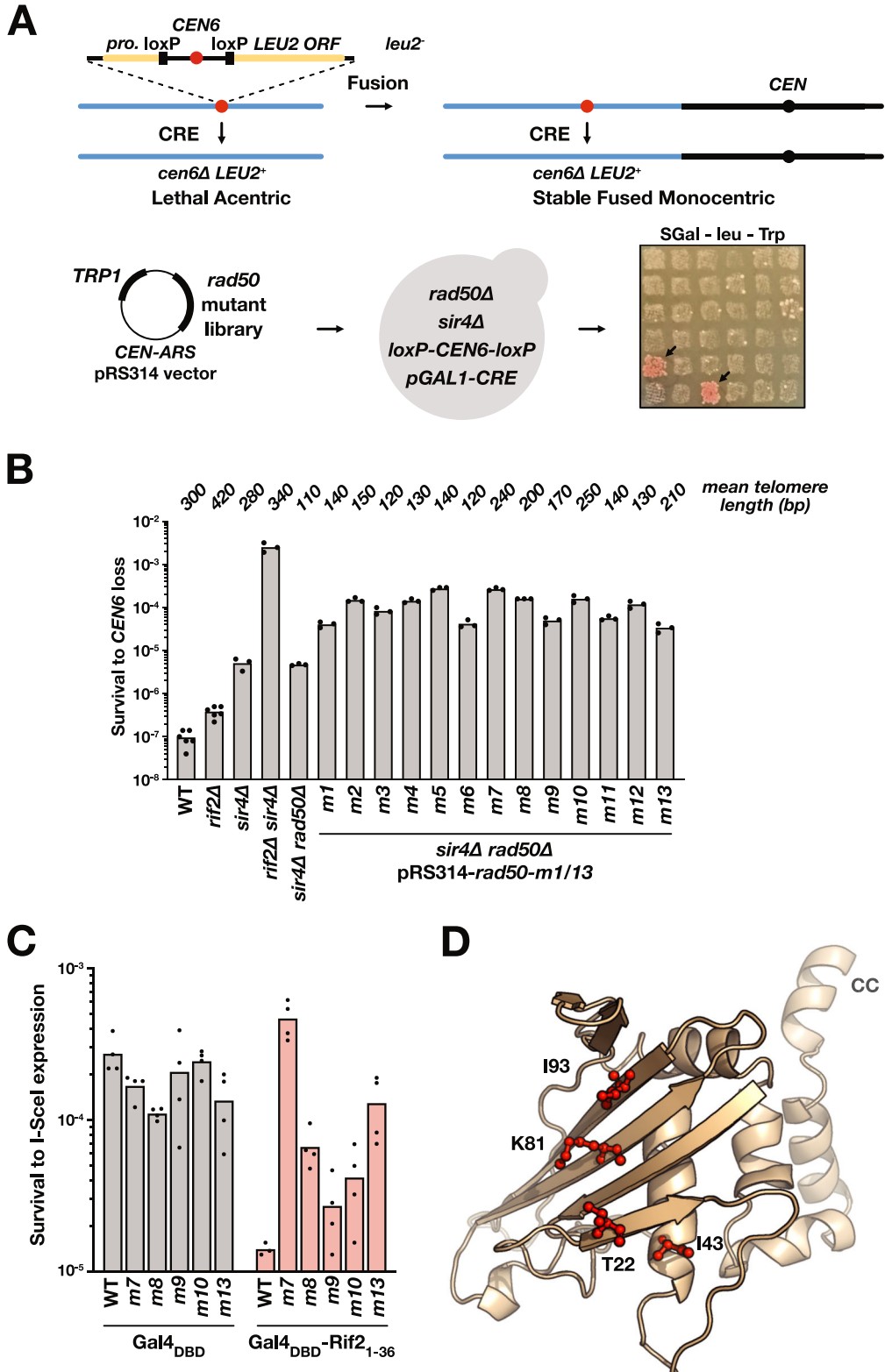

We expected two classes of mutants: (i) mutants whose increased fusion frequency is simply a consequence of a telomere shortening caused by a partial loss of function of MRX and (ii) mutants more specifically impacting Rif2 inhibition and better retaining MRX functions and the ability to elongate telomeres. Therefore, as a secondary screen, we determined the telomere length in the selected mutants. They display either very short telomeres (*m1* to *m6*, *m11*, *m12*) or mildly short telomeres (*m7* to *m10*, *m13*) (Fig. 5B, Supplementary Fig. 6A). We discarded the first group as unlikely to specifically resist Rif2. The candidates of the second group were assessed for their loss of sensitivity to Rif2 BAT using the I-SceI/Gal4 NHEJ assay. As shown in Fig. 5C, the 5 mutants maintain a frequency of survival to I-SceI expression close to the WT situation (Gal4DBD alone), indicating that NHEJ

**Fig. 5 Identification of Rif2-resistant Rad50 mutants through a genetic screen. A** Schematic representation of the chromosome fusion capture assay used to identity Rif2-resistant Rad50 mutants. The loss of chromosome 6 centromere (*CEN6*) generates a lethal acentric chromosome unless chromosome 6 fused to another chromosome. The Cre recombinase is expressed from a galactose-inducible promoter. Cre-induced *CEN6* loss generates a functional *LEU2* gene at the *CEN6* locus. The 5′ end of *RAD50* ORF (−105 to +996) was mutagenized by PCR using the Taq polymerase. The mutant library was transformed in cells lacking Sir4 and Rad50. 300 individual transformants were patched to saturation on rich medium prior to being replicated on synthetic medium with galactose (2%) and lacking leucine to identify clones with increased survival rate to *CEN6* loss. **B** Quantification of the survival to *CEN6* loss in the 13 *rad50* mutants identified in the screen (m1–m13). Cells were grown to saturation prior to plating on synthetic medium with galactose (2%) and lacking leucine. Colonies were counted after 5d at 30 °C. Means from independent cell cultures. Mean telomere length from Supplementary Fig. 5. **C** NHEJ inhibition by Rif2 BAT motif at I-SceI-induced broken ends in selected *rad50* mutants (single I-SceI site assay (Supplementary Fig. 2B)). Means from independent cell cultures. **D** Position of *rad50-m7* (K81) and *rad50-m13* (T22, I43, I93) mutated residues on a model structure of *S. cerevisiae* Rad50 N-terminal Head domain (1–189, obtained from *Chaetomium thermophilum* Rad50 structure as template[64] (PDB:5DAC)).

is still partially functional in these mutants. Targeting Rif2 BAT (1-36) to the break still represses NHEJ in the *m8*, *m9* and *m10* *rad50* clones, indicating that these mutants are still sensitive to BAT inhibition. In the *m7* and *m13* clones, survival to I-SceI expression is not reduced by BAT targeting. This result shows that these two mutants lead to NHEJ resistance to the BAT in this assay.

The *rad50-m7* allele carries three mutations, K81E, E242V and Q284L. The last two are within the N-terminal coiled-coil region. K81 is on a ß-sheet at the surface of Rad50 Head domain (Fig. 5D, model of *S. cerevisiae* Rad50ₐcc Head generated using *Chaetomium thermophilum* Rad50 structure as template[64] (PDB:5DAC 78% similarity/62% identity). The *m13* allele carries five mutations, three in the Head (T22I, I43T, I93T) and two in the coiled-coil region (I292M and I382V). T22 and I93 are near K81 and part of the same ß-sheet at the surface of the Head. I43 is within the domain below T22. The clustering of the selected mutations in the Head suggests that this Rad50 ß sheet surface is important for Rif2 BAT function.

**Identification of Rad50 residues essential for BAT function and interaction**. In parallel with the genetic screen described above, we used a structural modelling approach to search for Rad50 residues interacting with the BAT motif. Since BAT conformation is unknown, we used the flexible docking approach CABSdock[65] to perform the simulation search for the binding site, allowing full flexibility of the peptide and small fluctuations of the Rad50 backbone. To limit the combinatorial explosion due to the mainchain peptide flexibility, we used a short BAT sequence containing the most conserved amino acids (Rif2 residues 4 to 14, VDSDFAPIRRS; Fig. 1A). 10 000 structures were generated by CABSdock and the 1000 best scoring ones were clustered in 10 Rad50 NTD-BAT (4–14) peptide complexes. The clusters correspond to three interaction sites: one in the DNA-interaction site, one in the dimerisation site and one in the solvent-exposed ß-sheet where the mutations selected in the genetic screen are present. We therefore focused on this last cluster.

To validate this docking result, as well as to define more precisely the location of the BAT peptide, we used the Rad50ₐcc–BAT₄₋₁₄ complex models obtained with CABSdock to design mutations at several positions of the Rad50 ß-sheet (Fig. 6A). Mutations at positions K6, K81 and I93 abolish the Two-Hybrid interaction of Rad50 with Rif2 BAT (Fig. 6B, Supplementary Fig. 5). The same mutations preserve Rad50 interaction with Mre11. Mutations at surrounding positions on the ß sheet surface (S8, Q79, T95, N97 and Q115) maintain the Two-Hybrid interactions with Rif2 and Mre11 (S8L and T95L weakening the interaction with the BAT motif). In addition, purified Rad50ₐcc harbouring the K81E mutation fails to interact with Rif2 BAT in vitro (Fig. 6C; the F8A mutation within Rif2₁₋₆₀

also prevents this interaction). These results show a specific requirement of K6, K81 and I93 for the Rad50–BAT interaction.

Next, we tested the impact of the mutations on NHEJ inhibition by Rif2 BAT using the I-SceI/Gal4 assay. In the absence of the BAT motif at the broken end, the *rad50* mutants remain NHEJ proficient, with one exception Q115A (Fig. 6D). Targeting Rif2 BAT to the break represses NHEJ in cells with mutations at positions S8, Q79, T95 and N97 but not in cells with mutations at positions K6, K81 and I93. Mutations that prevent Rad50–BAT interaction also prevent NHEJ inhibition by the BAT motif. Altogether, this result, the independent result of the genetic screen and the docking model show that Rif2 BAT likely contacts the solvent-exposed ß-sheet of Rad50 N-terminal Head domain, residues K6, K81 and I93 playing a central role in this interaction. Furthermore, our results show that this interaction is essential to inhibit Rad50.

Since Rif2 BAT shortens telomeres[28] (Supplementary Figs. 2A, 3B, C), resistance to Rif2 BAT should cause telomere elongation. Among the numerous *rad50* mutants previously generated and studied, the *rad50S-K81I* allele results in longer telomeres[53,66,67]. This indicates that at least one residue essential for BAT function is also essential for proper telomere length homoeostasis. To further address this point, we tested the impact of mutations *K6A* and *K81E* on telomere length. As expected, both cause telomere elongation (Fig. 6E left panel, Supplementary Fig. 6B). Telomeres in *rad50-K6A* and *rad50-K81E* cells are not as long as in cells lacking Rif2 or bearing the *rif2-F8A* mutation, perhaps in part because the *rad50* mutations do not fully prevent Rif2 inhibition. In the absence of Rif2, the mutations *K6A* and *K81E* also lead to slightly different phenotypes. Telomeres are a little longer in *rif2Δ rad50-K6A* cells and a little shorter in *rif2Δ rad50-K81E* cells relative to *rif2Δ* cells. This suggests that the mutations moderately impact intrinsic Rad50 functions, in addition, to make it less sensitive to Rif2 (e.g. Tel1 activation[53,68]). In other words, they may not be perfect separation-of-function alleles. Rif1 is another repressor of telomere elongation that acts independently of Rif2[23,69–71]. As expected for mutations impacting Rif2 function, the *K6A* and *K81E* mutations still result in longer telomeres in cells lacking Rif1 (Fig. 6E, right panel, compare *rif1Δ* with *rif1Δ rad50-K6A* and *rif1Δ rad50-K81E*). Together with the previous finding that *rif2-F8A* and *rif2-R12A* BAT mutants elongate telomeres[28], these results show that telomere length homoeostasis relies in part on the Rad50–BAT interaction and its inhibitory function.

**Predicted impact of BAT binding on the MRX complex**. Next, we addressed the mechanism by which BAT binding on Rad50 Head inhibits MRX complex functions. First, we took advantage of the experimental results to build more accurate BAT-Rad50 models using the information-driven flexible docking approach HADDOCK[72]. In this method, ambiguous distance restraints that represent the spatial proximity between specific residues of the

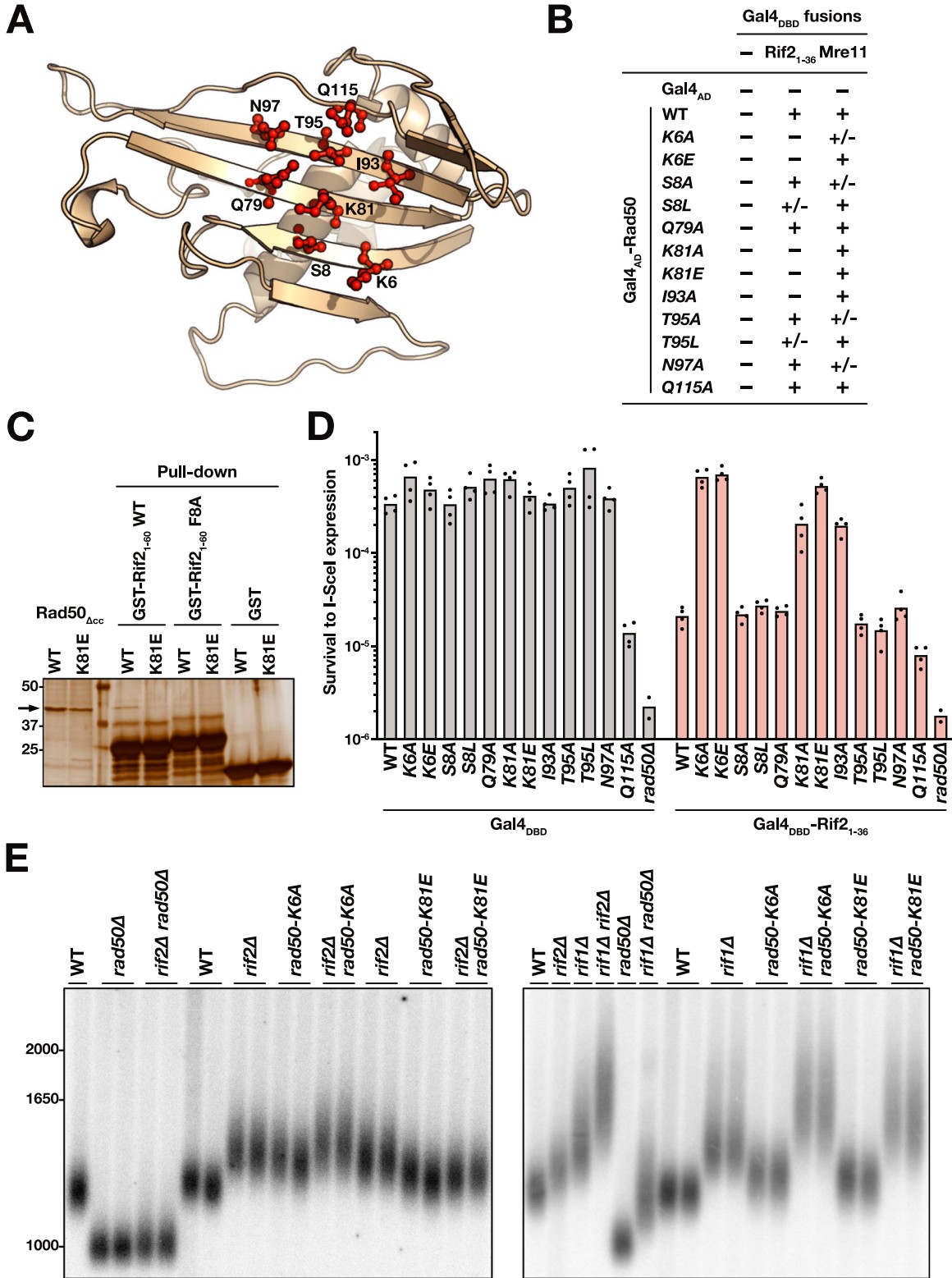

**Fig. 6 Identification of Rad50 residues essential for BAT function and interaction. A** Position of the eight residues (red) selected with CABSdock as potentially interacting with Rif2 BAT. **B** 2-Hybrids interactions between mutant full-length Rad50 and Rif2 N-terminal region or full-length Mre11. **C** GST pull-down interaction between Rif2 N-terminal region and Rad50 ATPase Head. Experiment reproduced three times. **D** NHEJ inhibition by Rif2 BAT motif at I-SceI-induced broken ends in rad50 mutants (single I-SceI site assay (Supplementary Fig. 2B)). Means from independent cell cultures. **E** Impact on telomere length of mutants rad50-K6A and rad50-K81E in WT cells and in cells lacking Rif1 or Rif2 (Southern blot, Y′ probe, XhoI digest). Experiment reproduced three times.

two partners are used to guide the docking. Residues K6, K81 and I93 of Rad50 and F8, R12, R13 and S14 of the $BAT_{4-14}$ peptide were selected as the interacting residues. The HADDOCK calculation led to four clusters. The ambiguous character of the restraints and the fully flexible treatment of the $BAT_{4-14}$ peptide explain the residual conformational spreading of the BAT peptide in these clusters. They all share the occupancy of a common region of the ß-sheet binding surface close to residue K6, K81 and I93. We then examined the consequences of this binding on the MRX complex structure.

How the MRX complex operates at a molecular level is not fully resolved yet but a recent cryo-EM study of *Escherichia coli* Rad50–Mre11$^{SbcC-SbcD}$ complex shows that it can adopt at least two states[61]. In an ATP-bound resting state, the Rad50 coiled-coils appear flexible and open. Upon DNA binding and consecutive ATP hydrolysis, the two coiled coils zip up into a rod and clamp DNA on the Rad50 Head. In this DNA-bound active state, Mre11 moves to the side of Rad50 to bind the DNA end. We used these structures to establish models of *S. cerevisiae* Rad50–Mre11 complex in the ATP-bound and DNA-bound states (Fig. 7A). To this end, we first built a model of *S. cerevisiae* Mre11 using Modeller[73] and *Chaetomium thermophilum* Mre11 structure as template[74] (PDB: 4YKE) (69% similarity/ 52% identity). To build the ATP-bound (resting state) and DNA-bound ScRad50-Mre11 models, we superimposed the ScRad50 and ScMre11 models on their *E. coli* homologues in the cryo-EM structures[61] (PDB:6SV6 and 6S85).

In the predicted ATP-bound state, K6, K81 and I93, the three Rad50 residues essential to BAT interaction, are exposed and located away from Mre11. However, the DNA-bound state brings helix 181–192 of Mre11 near the ß-sheet surface including residues K6, K81 and I93. Furthermore, in the docking models of the $BAT_{4-14}$ peptide on Rad50 obtained with HADDOCK, some BAT residues occupy the same region where helix 181–192 of Mre11 lies, as illustrated in Fig. 7A (bottom panel) (Supplementary Fig. 7). This suggests that the predicted BAT-bound state and DNA-bound state are not sterically compatible. In other words, the binding of the BAT peptide on Rad50 could prohibit the transition from the ATP-bound state to the DNA-bound state of the complex. This mechanism would explain how Rif2 BAT opposes MRX complex functions.

## Discussion

Our results lead us to propose a mechanism for MRX inhibition by Rif2 at telomeres (Fig. 7B). Like other SMC proteins, MRX likely uses ATP-driven conformational changes of its coiled-coil arms to scan and handle DNA[43,61,75,76]. MRX does not extrude loops but captures broken ends[60,61,77]. This end-capture is the first step that allows MRX to perform its functions in NHEJ repair, checkpoint activation, telomere elongation and 5′ end resection. Rad50 being a slow ATPase, both the ATP-bound resting state scanning for DNA ends and the DNA-bound active state are likely metastable[61]. At telomeres, Rap1 maintains Rif2 in close proximity to the DNA end[22]. We propose that, when an MRX complex approaches a telomere, the Rif2 BAT motif binds Rad50 ATPase Head in its ATP-bound resting state. This binding is favoured by Rif2 high local concentration at telomeres and Rad50 interaction surface accessibility in the resting state (Fig. 7 A). DNA binding stimulates ATP hydrolysis and the concomitant Mre11 move toward the DNA end. We propose that the BAT bound to Rad50 antagonises this last transition, plausibly by steric hindrance, and favours a rapid return to the more stable ATP-bound state. Unable to stably capture the telomere end, the MRX complex cannot proceed with the subsequent steps of NHEJ, Tel1-activation or 5′ resection. Once returned to the resting state,

the complex will diffuse away from Rif2 and the telomere end. In agreement with this last step and a dynamic low-affinity BAT–Rad50 interaction, Rif2 BAT does not stably retain MRX where it binds on DNA (Fig. 3B). Furthermore, Rif2 BAT only acts at short distances (Fig. 2C) and Rif2 overexpression is insufficient to inhibit NHEJ in the absence of recruitment to DNA ends (Supplementary Fig. 3A, B). The need for close spatial proximity between the inhibitor and its target is a key feature of the proposed mechanism. It ensures that MRX inhibition is restricted to telomeres where Rif2 binds and does not oppose MRX functions at other positions on chromosomes

In this model, Rif2 selectively destabilises one metastable state of the MRX complex, therefore accelerating the ATP-driven cycle between the two states. This futile cycle can explain the paradoxical ~2-fold stimulation of *S. cerevisiae* Rad50 ATPase activity by Rif2 in vitro[29,30]. Further supporting this interpretation, mutations in *E. coli* Rad50–Mre11 complex that challenge Rad50-Mre11 contacts specifically in the active DNA-bound state also stimulate Rad50 ATPase activity ~2-fold in vitro, likely again by causing faster ADP-to-ATP exchange within Rad50 ATP cycle[61]. Interestingly, *S. cerevisiae* Rad50 Q115 residue, whose mutation impedes NHEJ (Fig. 6D), virtually contacts Mre11 in the docking model of the DNA-bound complex (Supplementary Fig. 7B). Rad50 Q115 mutation may specifically challenge the DNA-bound active state, as we predict BAT presence on the adjacent residues would do. Non-exclusively, Rif2 binding may also directly impact the Rad50 ATP-bound state by stimulating ATP hydrolysis by an unknown mechanism[30].

Of the three residues identified as essential to interact with Rif2 BAT (Fig. 6), two (K6 and K81) also belong to a cluster of residues found mutated in the meiosis-defective *rad50S* alleles[61,78,79] (the *rad50-I93A* allele remains sporulation proficient, Supplementary Fig. 8A). *rad50S* mutants specifically impair Mre11-dependent 5′ resection due to a loss of interaction with the Sae2$^{CtIP}$ protein[80]. Thus, Sae2 and Rif2 BAT interaction interfaces are likely to partially overlap on Rad50 Head, suggesting that Rif2 could also antagonize Sae2 binding. Note that this last hypothesis cannot alone account for BAT functions since NHEJ repair, Tel1 activation and NHEJ inhibition by Rif2 BAT remain proficient in the absence of Sae2[30,81] (Supplementary Fig. 8B).

In addition to MRX inhibition by the BAT, the full-length Rif2 protein has other regulatory properties. For instance, it down-regulates DNA–RNA hybrids at telomeres, in part through the recruitment of RNaseH2[82]. Rif2 was also proposed to protect telomeres with the help of the chromatin remodeler Rpd3L[83,84]. Since RNaseH2 loss has no impact on telomere fusions (Supplementary Fig. 8C) and Rif2 BAT still inhibits NHEJ in cells lacking Rpd3L subunits Sin3 and Rxt2 (Supplementary Fig. 8B), these additional Rif2 functions are separable from the BAT functions.

MRX$^{MRN}$ and Tel1$^{ATM}$ inhibition is a conserved feature of telomeres in eukaryotes[3,7,15,17,42]. Is the mechanism of MRX$^{MRN}$ inhibition established by the BAT motif maintained in evolution? Rad50 ß-sheet and K6/K81 residues are evolutionarily stable in eukaryotes (e.g. R6/K81 in fission yeast and K6/R83 in human)[85–87], suggesting that this mechanism can be conserved. Since *K. lactis* Orc4 complements Rif2 loss in *S. cerevisiae* (Fig. 2G, Supplementary Fig. 3D), Orc4 could inhibit the MRX complex at telomeres in *Saccharomycetaceae* species lacking Rif2. In addition, MRX inhibition by Orc4 might have a function at replication origins, for instance protecting nascent strands from 5′ resection. Outside of the *Saccharomycetaceae* family, a conserved BAT motif is still present at Orc4 N-terminal end in some yeast species (e.g. *Candida auris*)[30] (Fig. 1A, Supplementary Fig. 1). It is followed by another conserved motif, which may correspond to a Rap1-binding motif, even though it does not

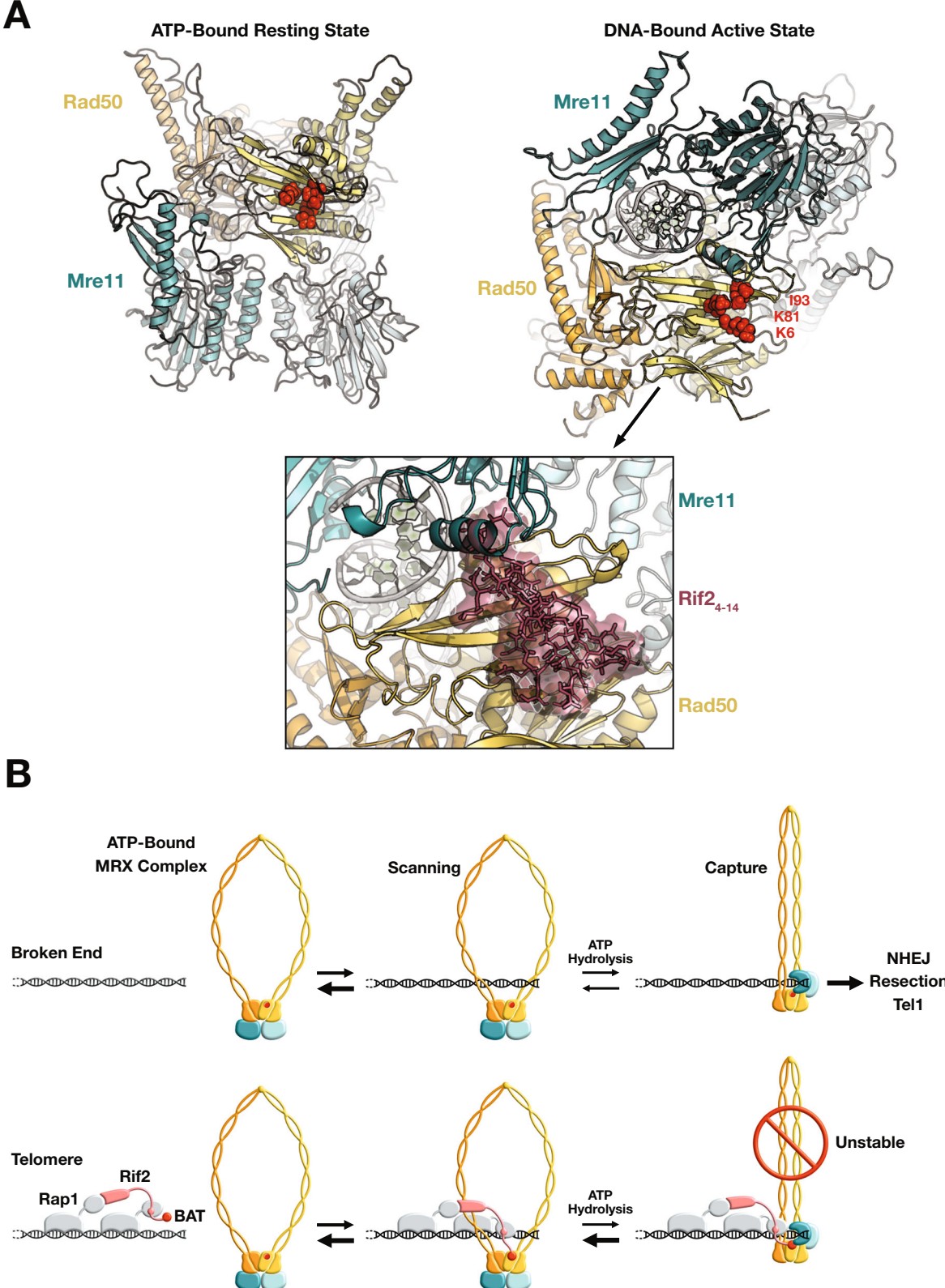

**Fig. 7 Predicted impact of BAT binding on the Rad50–Mre11 complex and model for MRX inhibition by Rif2 at telomeres. A** Models of *S. cerevisiae* Rad50–Mre11 complex in the ATP-bound resting state (left) and in the DNA-bound active state (right). Residues K6, K81 and I93 highlighted in red. Bottom panel: envelope of the BAT core peptide (Rif2 residues 4-14, in brown) belonging to the HADDOCK cluster 1 shown after superimposition of the Rad50 structures of the clusters on that of the DNA-bound Rad50–Mre11 complex model. **B** Model for telomere protection by Rif2 at telomeres (Rad50 orange/yellow, Mre11 teal/light blue, Rap1 grey, Rif2 pink, BAT and BAT interacting region on Rad50 red). For simplicity, Xrs2 and other factors present at broken ends and at telomeres are not represented.

perfectly match the sequence pattern defined from the *S. cerevisiae* Rif2 sequence.

In other eukaryotes, we could not find the BAT motif, at least corresponding to the strict pattern (D/E–F–X–X–Φ–R/K) (Supplementary Fig. 1). Searching for a small evolving linear motif, which can be embedded in highly degenerated sequences within intrinsically disordered segments, is difficult[88]. A direct screen for protein fragments interacting with Rad50 will likely be required to further address the conservation of this pathway, a candidate in mammals being the MRN-interacting iDDR motif within TRF2[7]. The yeast BAT core motif will also be a useful start to design artificial Rad50 inhibitors.

## Methods

**Strains, plasmids and molecular genetics.** Strains, plasmids, and primers used in this study are listed in Supplementary Data 1. Telomeres and telomere fusions were amplified using primers X2, Y'2 and polyG14[21,31]. The I-SceI assay used here was first described in[21]. Sites are inserted at the endogenous *URA3* locus. The I-SceI site inserted upstream of *URA3* is (I-SceI site bold, native sequence underlined): GTCCATAAGATCC**TAGGGATAACAGGGTAAT**AGATCTAAGCTTTT.

The I-SceI and Gal4 sites inserted downstream of *URA3* are (I-SceI site bold, native sequence and Gal4 binding sites underlined): TATTACCCTCGACGGAT CT**ATTACCCTGTTATCCCTA**GGATCGATCCTCTAGAGTCGGAGTACTGT CCTCCGAGCGGAGTACTGTCCTCCGAGCGGAGTACTGTCCTCCGAGCG GAGTACTGTCCTCCGAGCGGAGTACTGTCCTCCGAGGACCTGC AGGCAT GCAAGCTGATCCAATCTCGG

The peptide linker between Gal4$_{DBD(1-147)}$ and Rif2 fragments is -PELIPGDP GGGGGGGGGG.

To determine survival to I-SceI cleavage, cells were grown to saturation in synthetic medium lacking uracil with glucose (2%) ($1 \times 10^8$ cells/ml), diluted in water and spread on synthetic medium plates with galactose (2%). Colonies were counted after 3d at 30 °C.

To determine end resection, cells grown to OD 0.4 in synthetic glycerol lactate medium lacking uracil were arrested in G1 with $10^{-7}$ M α-factor (from a $10^{-3}$ M stock solution in ethanol) or in G2/M with 5 μg/mL nocodazole (from 1.5 mg/mL stock solution in DMSO) for 4 h. I-SceI expression was induced by galactose addition (2%). Genomic DNA was cut with StuI prior to gel electrophoresis. Southern blots were performed with a mix of two $^{32}$P-labelled probes hybridising *TIM9* and *ADE1*. Signal quantification was performed using a Typhoon 5 imager and the ImageQuant software.

Two-hybrid assays were performed using strain Y190 on synthetic medium plates with glucose (2%), adenine and His3 inhibitor 3-Amino-1,2,4-triazole (3AT) (25 and 50 mM). To screen for shorter Rad50 fragments interacting with the Rif2 BAT motif, plasmid pACT2-RAD50 was linearised by PvuII (situated in *RAD50* ORF midzone), partially digested by BAL-31 exonuclease and re-circularised with T4 DNA ligase. The library of random *RAD50* truncations was amplified in E. coli and transformed in yeast strain Y190 containing plasmid pRS414-Gal4$_{DBD}$-Rif2$_{1-36}$. Positive clones retaining the interaction were selected on plates with 50 mM 3AT. Plasmids recovered from yeast were sequenced. The shortest one encodes the Rad50$_{ΔCC}$ allele (Rad50$_{1-189-KILCY-1125-1312}$).

The 5′ end of *RAD50* ORF (−105 to +996; codons 1–332) was mutagenized by PCR using the Taq polymerase, a pRS314-RAD50 plasmid as a template and standard buffer condition (~$10^{4-5}$-fold amplification). The amplified fragment was reintegrated into PstI/StuI-digested pRS314-RAD50 by gap repair in yeast cells lacking Sir4 and Rad50.

**Protein purification and GST pull-down assay.** Rad50$_{ΔCC}$ and Rad50Δ$_{CC}$-K81E fused to His6-SUMO (N-terminal tag) were induced with 1 mM isopropyl-ß-D-thiogalactoside (IPTG) at 20 °C overnight into *E. coli* strain BL21 (DE3). All of the subsequent protein purification steps were carried out at 4 °C. Cells were harvested, suspended in lysis buffer (20 mM KPO4 pH7.8, 500 mM KCl, 1 mM DTT, 10% glycerol, 0.2% NP40, 1 mg/mL lysozyme, 1 mM 4-(2-aminoethyl) benzenesulphonyl fluoride (AEBSF), 10 mM benzamidine, 2 μM pepstatin) and disrupted by sonication. Extract was cleared by centrifugation at 186,000*g* for 1 hour at 4 °C and then incubated at 4 °C with NiNTA resin (QIAGEN) for 3 h. Mixture was poured into an Econo-Column® Chromatography column (BIO-RAD). After extensive washing of the resin with buffer A (20 mM KPO4 pH7.8, 150 mM KCl, 1 mM DTT, 10% glycerol, 0.2% NP40) complemented with 40 mM imidazole, protein was eluted with buffer A complemented with 400 mM imidazole. Fractions containing purified His-SUMO-Rad50$_{ΔCC}$ were pooled and applied to a PD10 column (GE Healthcare) equilibrated with buffer A to remove imidazole. Purified His-SUMO-Rad50Δ$_{CC}$ concentration was adjusted to 40 μM before storage at −80 °C.

GST-Rif2$_{1-60}$ and GST-Rif2$_{1-60}$-F8A were induced with 0.5 mM IPTG at 30 °C for 4 h into *E. coli* strain BL21 (DE3) and cells were disrupted by sonication into lysis buffer (50 mM Tris HCl [pH8@4 °C], 150 mM NaCl, 1 mM DTT, 1 mM EDTA, 0.2% NP40, 1 mg/mL lysozyme, 1 mM AEBSF, 10 mM benzamidine, 2 μM pepstatin). After centrifugation, the extract was incubated with GSH Sepharose

resin (GE Healthcare) overnight at 4 °C and then poured into an Econo-Column® Chromatography column (BIO-RAD). After extensive washing of the resin with buffer B (50 mM Tris HCl [pH8@4 °C], 150 mM NaCl, 1 mM DTT), proteins bound to the resin were eluted with buffer B complemented with 30 mM Glutathion. Fractions containing GST-protein were pooled and applied to a 2×1 ml Hitrap Heparine column (GE Healthcare) equilibrated with buffer C (50 mM Tris HCl [pH8@4 °C], 50 mM NaCl, 1 mM DTT, 1 mM EDTA). Protein was eluted with a 20 mL linear gradient of 0.05–0.4 M NaCl. Purified GST-ScRif2$_{1-60}$ was stored at −80 °C.

His-SUMO-Rad50$_{ΔCC}$ WT or His-SUMO-Rad50$_{ΔCC}$-K81E were cleaved with His-SUMO-Protease at a ratio of 1/20 (W/W) at 4 °C overnight. The mixtures were then incubated with NiNTA bead (BioRad) and Rad50$_{ΔCC}$ lacking the His-SUMO tag was recovered directly in the flow through. GST-Rif2$_{1-60}$ (10 μg), GST-Rif2$_{1-60}$-F8A (10 μg) or GST (10 μg) was immobilised on 10 μL Glutathione Sepharose 4B in 300 μl of buffer A (50 mM Tris HCl [pH8@4 °C], 150 mM NaCl, 1 mM DTT, 0.5 mM EDTA, 10% Glycerol) complemented with 2 mM MgCl2 and 25 units of benzonase for 60 minutes at 4 °C. The beads were collected by centrifugation, washed three times with 300 μl of buffer A. Rad50$_{ΔCC}$ (30 μg) was then added to the beads in 100 μL buffer A complemented with 2 mM MgCl2 and 25 units of benzonase) and incubation was pursued for 120 minutes at 4 °C with gentle agitation. The supernatant was removed and the beads were washed two times with 300 μL of buffer A. Proteins bound to the beads were then eluted by addition of 20 μL of 50 mM Tris-HCl [pH8@4 °C], 150 mM NaCl, 1 mM DTT, 30 mM glutathion. Proteins bound to the beads were resolved by 12% SDS-PAGE and detected by silver staining.

**Chromatin immunoprecipitation.** Cells grown to OD 0.4 in synthetic glycerol lactate medium lacking uracil were arrested in G1 with $10^{-7}$ M α-factor for 4 h. I-SceI expression was induced for 1.5 h by galactose addition (2%). Cells were crosslinked for 10 min with formaldehyde (1%) at 30 °C. Cell lysis and chromatin sonication were performed using a Bioruptor. Immunoprecipitation was performed using the 4A6 anti-myc antibody. Input and immunoprecipitated DNA concentrations were determined by qPCR.

**Homology modelling.** The N-terminal and C-terminal domains of *S. cerevisiae* Rad50 (Sc-Rad50) were modelled separately. Both domains were built by homology using Modeller 9.17[73]. We identified Rad50 from *Chaetomium thermophilum* (Ct-Rad50) as highly homologous to Sc-Rad50. The similarity between N-terminal domains of Sc-Rad50 (residue 2–189) and Ct-Rad50 is very high (62.8% identity/ 78% similarity). The sequence of the C-terminal domain of Sc-Rad50 (residue 1105–1312) is also highly similar to the one of Ct-Rad50 (residues 1105–1311; 61.2 % identity/73.7 % similarity). This made it possible to use the structure 5DAC[64] as a structural template for modelling both N- and C-terminal domains of *S. cerevisiae* Rad50. The best scoring models returned by Modeller were used for further investigations. Then, we build a model of Sc-Rad50 using the models obtained for its N- and C-terminal domains. The sequence used corresponds to the Rad50$_{ΔCC}$ fragment lacking the coiled-coil region.

A similar approach was used to model the catalytic domain of *S. cerevisiae* Mre11 (residue 1–415) involved in the interaction with Rad50. We used HMMER[89] in the bioinformatics toolkit[90] to identify a structural template in the protein database[91]. The X-ray structure 4YKE corresponding to the structure of Mre11 from *Chaetomium thermophilum* (Ct-Mre11) possesses a high sequence similarity with Mre11 from *S. cerevisiae* (Sc-Mre11) (52.2 % identity/69.7% similarity). Thus, 4YKE was selected as template to build the Sc-Mre11 model. 100 models were generated with Modeller 9.17 and the best scoring models were retained.

To build *S. cerevisiae* Rad50-Mre11 dimer models, we used the recently solved *E. coli* Rad50–Mre11 dimer complexes[61] in the resting state (structure 6S6V) and in the cutting state (structure 6S85). These EM structures allowed us to orient the two partners and to build the Rad50-Mre11 dimer in the resting state and the cutting one. As the sequence homology between Sc-Rad50 and Ec-Rad50 and between Ec-Mre11 and Sc-Mre11 is low (below 15%), the RMSD on the coordinates after superimposition of the whole structures is high (>2.5 Å). However, this permitted to obtain initial Sc-Rad50-Mre11 dimer models in the ATP-bound state and DNA-bound state. These initial complexes were refined by molecular dynamics in explicit water solvent. The starting structure was immersed in a cubic water box. The box was set using the Tcl plugin Solvate of VMD[92] so that the water layer around the solute (protein or DNA) was at least 12 Å. Then the system was neutralised with Na$^+$ or Cl$^-$ ions using the autoionize Tcl plugin of VMD. All MD calculations were carried out with the NAMD software[93] in the charmm36 forcefield[94]. The first step of the MD calculation consisted in a 1 ns restrained molecular dynamics simulation in the NPT ensemble. To this end, in addition to the standard energy terms (bonds, angles, dihedral angles, improper, van der Waals and electrostatic terms), we applied a harmonic potential (with constant $k = 1$ kcal mol$^{-1}$ Å$^{-2}$) restraining the CA atoms of the protein to their initial position while optimising sidechain atoms and water position. Then, a production step of 15 ns MD in the NPT ensemble was run without positional restraints. Frames were extracted from the last 5 ns MD trajectory and analysed. During MD, pressure control was achieved by a Nose-Hoover Langevin piston. MD

trajectories were analysed with VMD v 1.9.4. Modelling figures were produced with PyMol[95].

**Reporting summary**. Further information on research design is available in the Nature Research Reporting Summary linked to this article.

## Data availability

All relevant data are available from the authors without restriction. Raw images and quantitative data have been deposited in the "Mendeley repository" [https://doi.org/10.17632/8ghyn6n6x8.1].

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

## Acknowledgements

The authors thank Carol Greider for Rap1-Rif2$_{1-60}$ plasmid, Hélène Bordelet for pGBT9-RAD50 plasmid, Julie Soutourina for the 2H library, Elea Dizet (CIGEx platform) for plasmid construction and Stefano Mattarocci, Karine Dubrana, Laure Crabbé, Jean-Baptiste Boulé, Miguel Godinho Ferreira, Eric Coïc, Laurent Maloisel, Anna Campalans, Pablo Radicella, Teresa Teixeira, Hélène Bordelet, Alice Deshayes, Dan Throsby and Thomas M. Guérin, for fruitful discussions and suggestions. This work was supported by grants from *Agence Nationale de la Recherche* (ANR-14-CE10-021 DICENS, ANR-15-CE12-0007 DNA-Life, FRISBI ANR-10-INSB-0005), *Fondation ARC pour la Recherche sur le Cancer*, CEA Radiation biology programme and GGP CEA EDF programme. F.R.H. was supported by a PhD fellowship from ANR, a *Fondation ARC* young researcher grant and CEA.

## Author contributions

StM and F.R.H. conceived the study with help from all co-authors. F.R.H. performed the NHEJ, ChIP, Southern and Two-Hybrid assays. K.J. performed the telomere fusions experiments. S.P. and StM performed the genetic screen. D.B. created plasmids for protein purification and mutagenesis. J.D. and X.V. expressed and purified proteins and performed the GST pull-downs with help from SiM and S.Z.-J. P.C. performed molecular modelling, docking and MD simulations. I.C. performed the protein sequence alignements. StM and F.R.H. wrote the manuscript with editorial help from P.C., S.Z.J., J.B.C., I.C., M.H.L.D., D.B. and X.V.

## Competing interests

The authors declare no competing interests.
