## [Peer Review File · Nature Communications]

REVIEWER COMMENTS

Reviewer #1 (Remarks to the Author):

Suppression of the DNA damage response at telomeres is an important and conserved aspect of chromosome biology. The budding yeast Rif2 protein, which is recruited to telomeres by Rap1, suppresses the MRX-Tel1 pathway of telomere elongation. In this study, the authors identify the minimal "BAT" (blocks addition of telomere repeats) domain of Rif2 and critical residues within BAT for its activity in suppression of NHEJ and end resection, in addition to telomere elongation. Importantly, they show that BAT acts in cis in a distance dependent fashion, explaining why Rif2 inhibition of MRX is restricted to telomeres. Furthermore, they identify Rad50 as the BAT target of the MRX complex, consistent with recent studies from the Longhese and Burgers groups, and identify critical residues for interaction with the Rif2 BAT domain. By molecular modeling, the authors predict that interaction of the Rif2 BAT domain with Rad50 prohibits transition from the ATP-bound state to the DNA-bound state of the complex. This model nicely explains how BAT destabilizes MRX binding to Rap1-Rif2 bound ends and thus functions to suppress telomere addition and fusion.

Overall, this is an interesting and convincing study identifying Rad50 as the target for Rif2 inhibition of telomere addition, end resection and NHEJ. It is particularly interesting that some of the residues of Rad50 critical for BAT interaction are the same as those identified as rad50S alleles, which prevent Mre11 nuclease activation. It has been proposed that rad50S alleles are defective for a structural transition in the complex resulting from failed interaction with CDK-phosphorylated Sae2. It would be interesting to explore this further. Perhaps by testing if over-expression of Sae2 (or Sae2-S267E) competes with Rif2 causing telomere lengthening and/or fusion. Also, does the rad50-I93A allele confer a rad50S phenotype (meiosis defective, synthetic growth defect with sgs1)?

Minor comments:

Fig 4A and Fig S3: The interaction between BAT and Rad50 is weakened in mre11 and xrs2 mutants and it would be more appropriate to designate as +/- in Fig 4A. Fig 4B, D, moving the tables to a figure format has resulted in the text being misaligned.

Figure 5A is confusing and is not adequately explained in the text. First, although the basis for leu2 activation is in the Figure legend it would be helpful to show in the figure that deletion of CEN6 places leu2 next to a promoter, or describe the assay more fully in the text. Second, add TRP1 to CEN-ARS vector to make the need for -Leu -Trp selection clear to the reader. Third, there is no description of the rad50 mutagenesis in the Methods. Was the region subjected to PCR mutagenesis reconstituted in the full RAD50 ORF?

Fig S6 appears to be a compilation of unrelated experiments and the data are only mentioned in the Discussion. Relevant data, particularly the KIORC4 over-expression, should be reported in the Results section.

Reviewer #2 (Remarks to the Author):

In this manuscript, the authors characterized the role of the Rif2 BAT motif, known to limit telomere elongation and Tel1 activity. The authors showed that the BAT motif inhibits NHEJ and DSB resection by interacting with the Rad50 subunit of the MRN complex. They also identified residues on Rad50 that mediate Rad50-Rif2 interaction. They propose a model for Rif2 BAT function in the control of MRN activity.

The manuscript is interesting and most of the data are of high quality. However, some experiments are required to support the conclusions and strengthen the manuscript.

Specific points:

1. The authors show that Rif2 BAT blocks 5' resection (pag 7). This conclusion is not supported by experimental data as the assay they use does not measure ssDNA. The authors should use an assay that directly measures ssDNA.
2. The authors show by a two-hybrid approach that Rif2 N8A abolishes or reduces Rad50-Rif2 interaction. However, Hailemariam et al. showed biochemically that this mutation does not affect Rad50-Rif2 interaction. Since this is an important point in the paper, the authors should test biochemically whether the N8A mutation affects Rad50-Rif2 interaction.
3. Again, they use two-hybrid assay to identify Rad50 mutations that reduce/abolish Rad50-Rif2 interaction. At least one of the positive rad50 mutations should be tested biochemically to demonstrate that it reduces or abolishes Rad50-Rif2 interaction.
4. The length of telomeres in Fig 6 is very hard to explain. I don't understand why telomeres in rif2del rad50-K6A are longer than in rif2del. This effect cannot be explained by loss of Rad50 function as Rad50 is required to elongate telomeres.
5. Most of the experiments are based on the artificial targeting of Rif2-BAT to telomeres and DSBs. As this approach does not necessarily reflect the physiological function of Rif2 within the cell, which are the phenotypes of the rif2-N8A mutation with respect to telomere length and NHEJ?
6. In the model proposed by the authors resection and NHEJ depend on the MRX cutting state. Since these two processes are opposite, how can it be? The authors cannot ignore results from other papers that show that NHEJ and Tel1 activation depend on the ATP-bound state (for example Deshpande et al., 2014).
7. I am not an expert of homology modelling and docking. So, I leave this part to the other reviewers.

Reviewer #3 (Remarks to the Author):

The manuscript by Roisne-Hamelin et al. investigates the mechanism of Rif2 action at telomeres. By doing elegant and thorough experiments using genetics, molecular biology and some structural modelling, the authors demonstrate that the N-terminal BAT motif in Rif2 is sufficient to limit telomere elongation and Tel1 activity (as previously indicated) and to suppress telomere fusion. The BAT motif also hinders NHEJ and DNA resection when artificially recruited to engineered DNA double strand breaks, presumably all by directly disrupting the DNA end processing activity of the MRX complex and/or hindering MRX recruitment. Using Y2H assays the authors show that the BAT motifs (presumably directly) interacts with the Rad50 ATPase domain and identify point mutations in Rif2 that interfere with the interaction. Using a genetic screen, Rad50 mutations were isolated that enable telomere fusion, some of which are blocked in association with Rif2 as judged by Y2H. (Together with previous reports) this suggests that Rif2 directly associates via its BAT motif with Rad50 to regulate MRX.

A high-resolution structure of Rif2-BAT bound to MRX would be a great addition to the work, but it is not necessary for a basic understanding of the presented protection mechanism.

The experiments are well controlled and well presented, and the conclusions are robustly supported by the data. Some of the initial findings are consistent with results reported in Hailemariam et al., 2019a, Cassani et al. 2016, and Kaizer et al. 2015; but the manuscript goes well beyond these reports and the conclusions are actually different. Therefore, the manuscript is novel and important. It clarifies (some of) the roles of Rif2 at telomeres and clearly improves our understanding of the underlying mechanisms.

Suggested experiment:

The authors have chosen to place 5 UAS-Gal4 binding sites next to the DSB site (based on prior

constructions) for artificial recruitment of Rif2 BAT. Rad50 forms dimers. It would be interesting to determine whether one or both Rad50 protomers need to be bound to Rif2 for inhibition of NHEJ. The authors seem to have an ideal assay in hand to test this (by reducing the number of UAS-Gal4 sites). It would be a worthwhile addition to an already great manuscript.

Minor points:

While modelling might have been instrumental during the project, the conclusions seem to hold largely independently. The importance might thus be toned down accordingly in the abstract.

Fig. 1A. Orc4 sequences from WGS group of organisms could be included to indicate the 'loss of BAT and RBM motifs'. If so, then the alignment might have to be extended to the start of the AAA+ domain.

Fig. 2A. The scheme should indicate the role of Rif2 to facilitate easy understanding of the experiment.

Fig. 3B. A scheme with information on the locus would be helpful.

The bottom panel in Figure 7A is confusing. Better labeling and reduced complexity is needed to improve readability.

Discussion: Could the authors speculate/clarify as to how Rif2 might inhibit Mre11 activity while Sae2 (binding to a shared binding site) is needed for Mre11-dependent resection activity?

Point-by-point response to the reviewers' comments

We would like to thank the reviewers for their overall positive assessment and for their requests and suggestions to improve this work.

Reviewer #1

Remarks to the Author:

Suppression of the DNA damage response at telomeres is an important and conserved aspect of chromosome biology. The budding yeast Rif2 protein, which is recruited to telomeres by Rap1, suppresses the MRX-Tel1 pathway of telomere elongation. In this study, the authors identify the minimal "BAT" (blocks addition of telomere repeats) domain of Rif2 and critical residues within BAT for its activity in suppression of NHEJ and end resection, in addition to telomere elongation. Importantly, they show that BAT acts in cis in a distance dependent fashion, explaining why Rif2 inhibition of MRX is restricted to telomeres. Furthermore, they identify Rad50 as the BAT target of the MRX complex, consistent with recent studies from the Longhese and Burgers groups, and identify critical residues for interaction with the Rif2 BAT domain. By molecular modeling, the authors predict that interaction of the Rif2 BAT domain with Rad50 prohibits transition from the ATP-bound state to the DNA-bound state of the complex. This model nicely explains how BAT destabilizes MRX binding to Rap1-Rif2 bound ends and thus functions to suppress telomere addition and fusion.

Overall, this is an interesting and convincing study identifying Rad50 as the target for Rif2 inhibition of telomere addition, end resection and NHEJ. It is particularly interesting that some of the residues of Rad50 critical for BAT interaction are the same as those identified as rad50S alleles, which prevent Mre11 nuclease activation. It has been proposed that rad50S alleles are defective for a structural transition in the complex resulting from failed interaction with CDK-phosphorylated Sae2. It would be interesting to explore this further. Perhaps by testing if over-expression of Sae2 (or Sae2-S267E) competes with Rif2 causing telomere lengthening and/or fusion. Also, does the rad50-I93A allele confer a rad50S phenotype (meiosis defective, synthetic growth defect with sgs1)?

This is a very nice suggestion. To further address the intersection between Rad50 meiotic function and its interaction with Rif2 BAT, we asked whether our Rad50 mutants allow the sporulation of diploid cells. The rad50-K6E, -K81E and -K81A alleles are sporulation-defective, as expected. All the other tested alleles remain sporulation proficient. This includes the -K6A and -I93A mutants that disrupt the interaction with Rif2. This suggests that the Rif2 and Sae2 interaction surfaces on Rad50 only partially overlap. The new result is shown in Supplementary Figure S8A.

We also overexpressed Sae2 (2 μ plasmid, native promoter, full length or truncated (Δ N169), WT sequence or the phospho-mimetic S267E allele) and searched for a phenotype suggestive of a competition with Rif2 BAT on Rad50. Sae2 overexpression does not change telomere length nor the interaction between Rif2 BAT and Rad50 in a Two-Hybrid assay. Sae2 overexpression decreases NHEJ efficiency (I-SceI assay) but the residual activity remains sensitive to Rif2 BAT inhibition. It is difficult to interpret these negative results. For instance, the level of Sae2 overexpression may not be high enough to outcompete Rif2 since the latter is highly concentrated at telomeres. To further address this question will require more sophisticated approaches (e.g. targeting Sae2 to telomeres) and more direct readouts of the Sae2 interaction with the Rad50 β sheet.

Minor comments

Fig 4A and Fig S3: The interaction between BAT and Rad50 is weakened in mre11 and xrs2 mutants and it would be more appropriate to designate as +/- in Fig 4A. Fig 4B, D, moving the tables to a figure format has resulted in the text being misaligned.

Cells lacking MRX subunits have an intrinsic growth defect. This can explain the apparent weakening of the Two-Hybrid interaction. This is now pointed out in the legend of Figure 4. The alignment of the tables of Figure 4 is corrected.

Figure 5A is confusing and is not adequately explained in the text. First, although the basis for leu2 activation is in the Figure legend it would be helpful to show in the figure that deletion of CEN6 places leu2 next to a promoter, or describe the assay more fully in the text. Second, add TRP1 to CEN-ARS vector to make the need for -Leu -Trp selection clear to the reader. Third, there is no description of the rad50 mutagenesis in the Methods. Was the region subjected to PCR mutagenesis reconstituted in the full RAD50 ORF?

Changes in Figure 5A now highlight the basis for LEU2 activation and the TRP1 marker linked to the CEN-ARS vector. A description of the PCR mutagenesis was added to the legend of Figure 5A and in the Methods.

Fig S6 appears to be a compilation of unrelated experiments and the data are only mentioned in the Discussion. Relevant data, particularly the KlORC4 over-expression, should be reported in the Results section.

The data regarding Rif2 and KlOrc4 overexpression were moved to the Results section (Figure 2G and Supplementary Figure S3A/B/D).

Reviewer #2

Remarks to the Author:

In this manuscript, the authors characterized the role of the Rif2 BAT motif, known to limit telomere elongation and Tel1 activity. The authors showed that the BAT motif inhibits NHEJ and DSB resection by interacting with the Rad50 subunit of the MRN complex. They also identified residues on Rad50 that mediate Rad50-Rif2 interaction. They propose a model for Rif2 BAT function in the control of MRN activity.

The manuscript is interesting and most of the data are of high quality. However, some experiments are required to support the conclusions and strengthen the manuscript.

Specific points:

1. The authors show that Rif2 BAT blocks 5' resection (pag 7). This conclusion is not supported by experimental data as the assay they use does not measure ssDNA. The authors should use an assay that directly measures ssDNA.

The Southern Blot approach of Figure 3A and Supplementary Figure S4C offers a reliable way to assess broken end stability. To complement these results, we added a qPCR approach that

specifically amplifies the single strand DNA produced by 5' resection (Supplementary Figure S4B). It confirms that Rif2 BAT protects against broken end resection.

2. The authors show by a two-hybrid approach that Rif2 N8A abolishes or reduces Rad50-Rif2 interaction. However, Hailemariam et al. showed biochemically that this mutation does not affect Rad50-Rif2 interaction. Since this is an important point in the paper, the authors should test biochemically whether the N8A mutation affects Rad50-Rif2 interaction.

3. Again, they use two-hybrid assay to identify Rad50 mutations that reduce/abolish Rad50-Rif2 interaction. At least one of the positive rad50 mutations should be tested biochemically to demonstrate that it reduces or abolishes Rad50-Rif2 interaction.

Thanks for asking us to address the interaction between Rad50 Head domains and Rif2 *in vitro*. We purified the Rad50 Δ CC form lacking the coiled-coil region and GST-Rif2₁₋₆₀ fusions expressed in *E. coli* cells. In a GST pull-down assay, the two peptides interact (Figure 4E). The Rif2 F8A mutation and the Rad50 K81E mutation prevent this *in vitro* interaction (Figure 6C). These new data reinforce the Two-Hybrid results and confirm that the interaction is direct and specific.

4. The length of telomeres in Fig 6 is very hard to explain. I don't understand why telomeres in rif2del rad50-K6A are longer than in rif2del. This effect cannot be explained by loss of Rad50 function as Rad50 is required to elongate telomeres.

To address this point further, we combined the *rad50-K6A* and *-K81E* alleles with the *rif2-F8A* allele. We observed the same partial epistasis between these alleles (new data in Supplementary Figure S6C). The differences in telomere length between the single *rif2* mutants (Δ or F8A) and the double *rif2 rad50* mutants are small but significant. This suggests that the *rad50* mutations moderately impact intrinsic MRX functions in addition to make it less sensitive to Rif2 (e.g. hyper- or hypo-activation of Tel1).

5. Most of the experiments are based on the artificial targeting of Rif2-BAT to telomeres and DSBs. As this approach does not necessarily reflect the physiological function of Rif2 within the cell, which are the phenotypes of the rif2-N8A mutation with respect to telomere length and NHEJ?

The *rif2-F8A* mutation at the endogenous *RIF2* locus elongates telomeres (Supplementary Figure S3C) and exposes them to NHEJ (Figure 2F).

6. In the model proposed by the authors resection and NHEJ depend on the MRX cutting state. Since these two processes are opposite, how can it be? The authors cannot ignore results from other papers that show that NHEJ and Tel1 activation depend on the ATP-bound state (for example Deshpande et al., 2014).

Deshpande et al. 2014 is an important paper to understand how Rad50 acts. It is now cited in the introduction and the discussion. One of its key results is that end-tethering by MR requires ATP hydrolysis *in vitro* (Figure 5A, EMBOJ 2014 33:482). This is consistent with our model that the alternation between the ATP-bound state and the DNA-bound state is essential to all MRX functions at DNA ends, including NHEJ. In eukaryotes, Sae2-dependent resection by MRX is initiated slowly, giving time for multiple attempts at NHEJ repair. Therefore, the same state can carry out both functions. The mechanism holding the brake on DNA cleavage and ensuring the slow initiation of 5' resection remains to be deciphered.

7. I am not an expert of homology modelling and docking. So, I leave this part to the other reviewers.

Reviewer #3

Remarks to the Author:

The manuscript by Roisne-Hamelin et al. investigates the mechanism of Rif2 action at telomeres. By doing elegant and thorough experiments using genetics, molecular biology and some structural modelling, the authors demonstrate that the N-terminal BAT motif in Rif2 is sufficient to limit telomere elongation and Tel1 activity (as previously indicated) and to suppress telomere fusion. The BAT motif also hinders NHEJ and DNA resection when artificially recruited to engineered DNA double strand breaks, presumably all by directly disrupting the DNA end processing activity of the MRX complex and/or hindering MRX recruitment. Using Y2H assays the authors show that the BAT motifs (presumably directly) interacts with the Rad50 ATPase domain and identify point mutations in Rif2 that interfere with the interaction. Using a genetic screen, Rad50 mutations were isolated that enable telomere fusion, some of which are blocked in association with Rif2 as judged by Y2H. (Together with previous reports) this suggests that Rif2 directly associates via its BAT motif with Rad50 to regulate MRX. A high-resolution structure of Rif2-BAT bound to MRX would be a great addition to the work, but it is not necessary for a basic understanding of the presented protection mechanism. The experiments are well controlled and well presented, and the conclusions are robustly supported by the data. Some of the initial findings are consistent with results reported in Hailemariam et al., 2019a, Cassani et al. 2016, and Kaizer et al. 2015; but the manuscript goes well beyond these reports and the conclusions are actually different. Therefore, the manuscript is novel and important. It clarifies (some of) the roles of Rif2 at telomeres and clearly improves our understanding of the underlying mechanisms.

Suggested experiment:

The authors have chosen to place 5 UAS-Gal4 binding sites next to the DSB site (based on prior constructions) for artificial recruitment of Rif2 BAT. Rad50 forms dimers. It would be interesting to determine whether one or both Rad50 protomers need to be bound to Rif2 for inhibition of NHEJ. The authors seem to have an ideal assay in hand to test this (by reducing the number of UAS-Gal4 sites). It would be a worthwhile addition to an already great manuscript.

This is a very nice suggestion. Strikingly, the targeting of a single Gal4_{DBD}-Rif2₁₋₃₆ dimer inhibits NHEJ (1 Gal4 site in Figure 2B and Supplementary Figure S2B), showing that one or two BAT molecules are sufficient to act. Whether the binding of a single Rad50 protomer can inhibit the complex remains open since the Gal4-based approach does not allow the targeting of a single molecule. We will need to develop other approaches.

Minor points:

While modelling might have been instrumental during the project, the conclusions seem to hold largely independently. The importance might thus be toned down accordingly in the abstract.

The docking model contributed at two levels: (i) to explore by mutagenesis the BAT-interacting surface on Rad50 and (ii) to predict the impact of BAT binding on MRX. We thought that the abstract should express this.

Fig. 1A. Orc4 sequences from WGS group of organisms could be included to indicate the 'loss of BAT and RBM motifs'. If so, then the alignment might have to extended to the start of the AAA+ domain.

An alignment of the full-length Rif2 and Orc4 proteins is now provided in Supplementary Figure S1.

Fig. 2A. The scheme should indicate the role of Rif2 to facilitate easy understanding of the experiment.

At this position in the figure, it seemed difficult to indicate what Rif2 does prior to the results.

Fig. 3B. A scheme with information on the locus would be helpful.

A scheme of the locus surrounding the I-SceI site is provided in Supplementary Figure S4A. It now includes the position of the primers used for the CHIP.

The bottom panel in Figure 7A is confusing. Better labeling and reduced complexity is needed to improve readability.

A coloured labelling of the bottom panel in Figure 7A has been added to improve readability.

Discussion: Could the authors speculate/clarify as to how Rif2 might inhibit Mre11 activity while Sae2 (binding to a shared binding site) is needed for Mre11-dependent resection activity?

Rad50 I93 is not required for meiosis (Supplementary Fig. S8A). In the DNA-bound state of MRX, I93 is closer to Mre11 than K81 and K6 (Supplementary Fig. S7B). We can speculate that part of Rif2 specificity is its interaction with I93 and the steric clash with Mre11 to which it contributes. Sae2, by not interacting with I93, would not oppose Mre11.

REVIEWERS' COMMENTS

Reviewer #1 (Remarks to the Author):

The authors have addressed all of my concerns. Their findings provide new insight into how the interaction between Rif2 and Rad50 inhibits MRX functions at telomeres and will be of broad interest to the chromosome biology field.

Reviewer #2 (Remarks to the Author):

I am satisfied with the authors' responses to my suggestions and comments.

Reviewer #3 (Remarks to the Author):

The authors have improved the manuscript during the revisions by addressing the comments raised by the reviewers. Several added experiments provide additional support for the main conclusions of the work. The manuscript should now be ready for publication.